# Pharmacological and Chemical Analysis of *Bauhinia divaricata* L. Using an In Vitro Antiadipogenic Model

**DOI:** 10.3390/plants12223799

**Published:** 2023-11-08

**Authors:** Ana Laura Islas-Garduño, Ofelia Romero-Cerecero, Antonio Ruperto Jiménez-Aparicio, Jaime Tortoriello, Rosa Mariana Montiel-Ruiz, Manases González-Cortazar, Alejandro Zamilpa

**Affiliations:** 1Centro de Desarrollo de Productos Bióticos, Instituto Politécnico Nacional, Yautepec 62739, Mexico; alislasg92@gmail.com; 2Centro de Investigación Biomédica del Sur, Instituto Mexicano del Seguro Social, Xochitepec 62790, Mexico; orcerecero@yahoo.com.mx (O.R.-C.); jtortora2@yahoo.es (J.T.); montielrmariana@gmail.com (R.M.M.-R.); gmanases@hotmail.com (M.G.-C.)

**Keywords:** *Bauhinia divaricata*, ethyl acetate, adipogenesis, 3T3-L1, obesity, adipose tissue

## Abstract

Obesity is characterized by an excessive and abnormal accumulation of fat. According to the 2022 National Health and Nutrition Survey, in Mexico, the prevalence of overweight and obesity—diagnosed if one’s body mass index (BMI) was ≥25 kg/m^2^—in adults was 75.2%. A strong association between the amount of visceral fat and diseases such as diabetes mellitus type II has been recognized. Species of the Bauhinia genus have lipid-lowering and antidiabetic properties. The aim of this work was to evaluate the lipolytic and antiadipogenic activity of *Bauhinia divaricata* L. in 3T3-L1 cells and to identify the major compounds in the bioactive treatments. The extraction of aerial parts allowed us to obtain hexanic (BdHex), ethyl acetate (BdEAc), and hydroalcoholic (BdHA) extracts. Lipid levels were measured in 3T3-L1 cells differentiated into adipocytes. Our evaluation of cell viability identified an IC_50_ > 1000 μg/mL in all the extracts, and our evaluation of the antiadipogenic activity indicated that there was a significant reduction (*p* < 0.001) in the accumulation of lipids with hydroalcoholic (60%) and ethyl acetate (75%) extracts of *B. divaricate* compared with metformin at 30 mM (65%). The major compounds identified in these extracts were as follows: triacetin (**1**), 2,3-dihydroxypropyl acetate (**2**), (3E)-2-methyl-4-(1,3,3-trimethyl-7-oxabicyclo[4.1.0]hept-2-yl)-3-buten-2-ol (**3**), 2,5-dihydroxyphenylacetic acid (**4**), (3R)-3-hydroxydodecanoic acid (**5**), kaempferol-3-O-rhamnoside (**6**), and quercetin 3-O-rhamnoside (**7**). Some of these naturally occurring compounds have been related to the anti-obesity effects of other medicinal plants; therefore, these compounds isolated from *B. divaricata* could be responsible for inhibiting the differentiation process from preadipocytes to mature adipocytes.

## 1. Introduction

The prevalence of obesity has grown exponentially over the last two decades, and it has been labeled a pandemic-like problem. Currently, its prevalence continues to increase, and patients suffering from this illness are at risk of developing chronic diseases [1]. Mexico and the United States occupy the tops places in terms of the prevalence of obesity among adults. Particularly, in Mexico, in the year 2018, 75.2% of the population was found to be overweight or obese; of those with obesity, more than 75% had abdominal obesity, which is related to the development of chronic pathologies in adults such as type II diabetes mellitus, high blood pressure, and cancer. In 2021, the prevalence of overweight and obesity (BMI ≥ 25 kg/m^2^) was 75.0% in women and 69.6% in men [2,3,4]. There is a strong association between abdominal fat and cardiovascular and metabolic disease. This same association, according to the World Health Organization (WHO), increases the risk associated with health to a severe level when the individual’s BMI is higher [5]. In addition to the risks mentioned above, there is also a high risk of death, which is why, in recent years, studies on obesity and the mechanisms that are involved in its development have gained important relevance [6]. The WHO has classified obesity as the epidemic of the 21st century; genetic and environmental factors determine this metabolic disorder, which is characterized by an excessive accumulation of body fat above the expected amount [7]. The accumulation of fat in the abdominal region or central obesity is of special interest, considering the measurement taken on the iliac crest is ≥102 cm in men and ≥88 cm in women since obesity is directly related to the risk of presenting cardiovascular, metabolic complications, and insulin resistance [8,9]. Said fatty tissue alters its function qualitatively and quantitatively in its capacity to store fat, and inflammation of the same tissue linked to metabolic disorders can occur [10]. The treatment of obesity is multidisciplinary and involves the use of drugs such as phentermine, topiramate, and orlistat, among others, all of which have adverse effects on the consumer [11,12]. The use of plant species for weight control has also been reported, and species with a history of lipid and blood glucose control can be sourced from the genus *Bauhinia*. Among the most outstanding species of the *Bauhinia* genus is *B. divaricata* L., which is one of the most common species found in the Yucatan Peninsula and is recognized as one of the five wild species of the native representative Cercideae tribe [13,14]; it can also be found in other states of Mexico [15]. *B. divaricata* has an ethnomedical history of being used as a lipid-lowering, anti-inflammatory, antimicrobial, and anticoagulant treatment for diabetes. Studies focusing on the species *B. divaricata* have reported on its glycolipids, glycosyl-steroids, lactones, quinines, tannins, and terpenoids [16,17]. Due to the lack of specific studies for the treatment of obesity using the *Bauhinia* species, the objective of this work was to evaluate the antiadipogenic activity of *B. divaricata* in an in vitro model of obesity.

## 2. Results

### 2.1. Cell Viability Determined Using an MTT Assay in 3T3-1 Cells Treated with the Extracts Obtained from B. divaricata

The BdHex, BdEAc, and BdHA extracts have proven themselves not to be toxic for these cells at different concentrations (31.2–1000 μg/mL), resulting in a mean inhibitory concentration (IC_50_) greater than 1000 μg/mL (Table 1). Therefore, it was decided to evaluate concentrations of 100, 200, and 300 μg/mL of each of the extracts.

### 2.2. Lipid Quantification with Oily Red Assay in the Three Extracts Evaluated from Bauhinia divaricata on Days 5, 7, and 9 of Differentiation in 3T3-L1 Cells

The extracts that managed to control the total lipid levels were BdHex, BdEAc, and BdHA (Figure 1). The results were statistically significant with respect to the vehicles for the three concentrations. The greatest decrease was reached in the concentration of 200 μg/mL of the BdHA of *B. divaricata* on day 9 of differentiation.

### 2.3. Chromatographic Fingerprints of Integrated Extracts

The BdHA extracts and the BdEAc fraction presented greater antiadipogenic activity, which is why they were analyzed using high-performance liquid chromatography (HPLC) to identify their chemical contents (Figure 2). The compounds identified for BdEAc of *B. divaricata* were as follows: flavonols and flavone; the compounds identified for BdHA of this same species were as follows: flavonol, flavone, and polyphenol.

### 2.4. Lipid Quantification Using Oily Red Assay of BdEAc on Days 5, 7, and 9 of Differentiation in 3T3-L1 Cells

The fractionation of BdEAc allowed us to obtain the TI–TIV treatments, which were evaluated using the method for adipogenesis to determine the concentration of the total lipids (Figure 3a–d). All the treatments showed an effect during the 9 days, but both TIII and TIV showed a decrease in concentration from day 5, which was not observed in the other two treatments (T1 and TII). This effect was more evident in TIII at its lowest concentration.

### 2.5. Chemical Characterization of TIII from BdEAc

Since treatment TIII was the one that showed the best reducing activity in terms of the number of total lipids, its chemical separation was carried out using open-column chromatography monitored with thin-layer chromatography and HPLC (Figure 4a). The main compounds that appeared at minutes 7.008 (**1**) and 10.147 (**6**) were observed; these compounds were of interest because they were the majority compounds of the mixture (Figure 4b).

The BdTIIIEAc was obtained by separation using open-column chromatography and monitored by using thin-layer chromatography. A total of 29 fractions were collected. In fraction TIII-8, we found the majority compound (Figure 4c).

In addition, TIII-B was separated using open-column chromatography, where a total of 36 fractions were obtained, identifying the presence of compounds of interest in fraction TIIIB-20 (Figure 4d).

After the chemical fractionation of BdTIIIEAc, nuclear magnetic resonance and ultra-performance liquid chromatography were carried out to identify the chemical structure of the compounds responsible for the in vitro activity as well as their molecular weight. According to the results derived from our analysis of the nuclear magnetic resonance (NMR) data (Table 2) and comparing with data in the literature [18,19], this compound corresponds to triacetin (**1**). ESI-MS data indicated positive ions of **1** at m/z 241.04 [M+Na]^+^, 219.01 [M+H]^+^, and 159 [M-AcO]^+^.

Compound (**6**) was isolated as a yellow powder. In the UV light spectrum, the compound showed a λmax 262 and 343.4 nm, which is characteristic of a flavonol. The ^1^H NMR spectra showed two systems: an aromatic AB system, [δ 6.21 (1H, d, 1.9 Hz, H-6) and 6.38 (1H, d, 1.9 Hz, H-8)], that signals other aromatic ABA’B’ systems, such as [δ 7.78 (1H, d, 8.7 Hz, H-2′, H-6′), and 6.95 (1H, d, 8.7 Hz, H-3′ and H-5′)], which indicates a flavonol nucleus called kaempferol. One anomeric proton at δ 5.39 (1H, d, 1.4) was assigned to H-1″ correlates in COSY with a signal at δ 4.23 (1H, dd, 1.4, 3.3), which was assigned to H-2″. Regarding the sugar portion, the evaluation of spin–spin coupling, and chemical changes enabled the identification of Rhamnopyranose as a sugar residue. The position of sugar with kaempferol was unambiguously defined at C-3(δ 134.7) with the long-range correlation (HMBC) with the proton of H-1″ at (δ 5.3, d, 1.4 Hz) kaempferol-3-O-rhamnoside (**6**). In ESI-MS, it showed a positive ion at 433.11 [M+H]^+^, and its molecular formula is C_21_H_21_O_10_.

On the basis of this information and the direct comparison of spectroscopic data (see Table 3) described in the literature, this natural product was identified as kaempferol-3-O-rhamnoside (**6**); see Figure 5.

### 2.6. GC-MS of Treatment TIII-8

The chemical composition of TIII-8 was complemented using gas chromatography–mass spectrometry. Major compounds are described in Table 4.

### 2.7. Chemical Characterization of TIV from BdEAc

A comparison of the HPLC chromatogram of the BdTIVEAc (Figure 6a) fraction with TlV (Figure 6b) showed the presence of a major compound at minute 9.53 in TIV-13 (Figure 6c). The mass spectrometry (ESI-MS) results indicated positive ions at *m*/*z* 301.08 [M-Rhamnoside]^+^ and 449.12 [M+H]^+^, and their molecular formulas are C_15_H_9_O_7_ and C_21_H_22_O_11_, respectively. A comparison of these data with data from the literature [20] did not indicate that this compound corresponds to quercetin 3-O-rhamnoside (**7**); see Figure 7.

## 3. Discussion

Among the different species of the Bauhinia genus, it has been reported that the leaves and flowers of the *Bauhinia forficata* subsp. prurinosa are used in countries like Chile in the form of an infusion for glycemic control and that they have pharmacological effects [21]. For the present work, *B. divaricata* (aerial parts) was selected because reports have shown that this species has a high content of secondary metabolites and it was possible to identify that the aqueous extract of the leaves shows hypoglycemic activity. The *Bauhinia* genus is known in different Latin American countries and has been especially studied in Brazil, where it was used to treat infections, pain, and diabetes [22].

Previous work carried out by Herrera-Ruiz et al. (2019) led to the obtention of a hydroalcoholic extract of *Bauhinia blakeana* with a yield of 7.8%, which is similar to our results of a yield of 7.30% in the hydroalcoholic extract of *B. divaricata* [23].

In the evaluation of the effect produced by the hydroalcoholic extract of *B. divaricata* on the differentiation of 3T3-L1 cells, it was possible to identify the induction of a remarkable morphological change that manifests itself from the third day, mainly in its size and taking a shape, frank round on day 9 of differentiation. The cells lost the typical stellate morphology of the fibroblast and became multiple lipid spheres that also increased in size as the differentiation process progressed. Clavijo et al. previously reported the same for the 3T3-L1 cell line (2007) [24].

Regarding the evaluation of viability, the results of this work show that the extracts from *B. divaricata* did not present a cytotoxic effect on the 3T3-L1 cell line. However, there are studies, such as that of Sharma et al. (2019) [25], wherein the methanolic extract of *Bauhinia variegata* L. bark showed a cytotoxic effect on glioma C-6 cell lines, MCF-7 breast cancer cells, and colon cancer cells. HCT-15. Similarly, *Bauhinia purpurea* was shown to have cytotoxic activity in the KB and BC cell lines, with which it reached an IC50 of 10.5 to 72.3 μM [26].

Studying the extracts in the in vitro model made it possible to identify the extracts with the highest activity among the above and select one of them to carry out chemical separation to identify the major compounds in the bioactive treatments. The ethyl acetate extract of the species *B. divaricata* was subjected to chemical separation tests, and from these tests, it was possible to obtain 118 fractions, which were collected in four treatments (I, II, III, and IV), of which the one with the highest activity was treatment III. In fraction TIII-8, the compound triacetin was identified, and in TIIIB-20, the compound identified was kaempferol-3-O-rhamnoside. As far as triacetin is concerned, there is no history of it being identified in the species, and it has only been used as a food additive. Regarding kaempferol-3-O-rhamnoside, in 2015, it was reported in the species *Bauhinia malabárica* in a study carried out in Egypt as afzelin, as well as in the compilation of different species of the genus *Bauhinia*, among which was *Bauhinia variegate*, which was reported to have a molecular weight of 431.09 g/mol. These results are similar to what was identified in this study, where ultra-high resolution chromatography and masses of molecular weight were used to identify kaempferol-3-O-rhamnoside with a molecular weight of 433.12 g/mol [27].

Kaempferol-3-O-rhamnoside belongs to the most abundant group of chemical compounds in the *Bauhinia* genus. The kaempferol-3-O-rhamnoside found in the *Schima wallichii Korth* species native to Asia has shown that it manages to reduce lipid peroxidation, which is related to obesity if the patient has a higher than usual body mass index (determined by the amount of abdominal fat that the adipocyte, which is compressed within the abdomen), is capable of releasing, a greater amount of reactive oxygen species and proinflammatory cytokines, and a condition for the development of lipid peroxidation [28]. In a study carried out with quercetin 3-O-rhamnoside, its activity as an inhibitor of lipid peroxidation in 3T3-L1 cells of the same cell type used in our work was recognized [29]. This could indicate that the effect observed in the quantification of total lipids is related to the antioxidant effect of the polyphenolic compounds between others. 2,5-dihydroxyphenylacetic acid (4) has a history of identification in *Entada phaseoloides* (L.), a medicinal species used in China to treat problems of hemostasis and detoxification and as a treatment for diabetes mellitus. It is also anti-inflammatory and has recently been found to have anti-HIV activity [30].

In addition, it is important to highlight that several Bauhinia species are native to Mexico, and it is possible to find them in abundance in different states of Mexico [31]. This is extremely important due to the obesity problems that this country suffers from.

## 4. Materials and Methods

### 4.1. Plant Materials

The aerial parts of *B. divaricata* were collected in the state of Quintana Roo in January. One sample was kept for identification in the herbarium of the Mexican Social Security Institute (*B. divaricata* registry no. 16809).

### 4.2. Extraction

The aerial parts of *Bauhinia divaricata* were dried in an oven for 48 h and then ground in an industrial mill (Pulvex, Mexico City, Mexico) until a particle size of 4 mm was obtained. The dry plant material (1798 kg) was subjected to successive maceration with solvents of different polarities, namely hexane (4 L), ethyl acetate (4 L), and ethanol/water (60:40, 4 L). Each extract was filtered (whatman No. 4) and concentrated in a rotary evaporator (Heidolph G3, Schwabach, Germany) until dry, and subsequently, the extracts were lyophilized to give them *n*-hexane (BdHex), ethyl acetate (BdEAc), and hydroalcoholic BdHA), respectively.

### 4.3. Cell Line Differentiation: 3T3-L1

The 3T3-L1 (CRL-173) ATCC (American Type Culture Collection) cell line was used for this assay. DMEM (Dulbecco’s Modified Eagle Medium) medium supplemented with 10% FBS (Fetal Bovine Serum), essential amino acids, L-glutamine, sodium bicarbonate, and antifungal antibiotic (In vitro, S.A., Mexico City, Mexico) was used. The cells were cultivated in 75 cm^2^ flasks at 37 °C and with 5% CO_2_; the medium was changed every third day until it reached 80% confluence (observed under a microscope—Olympus CK2). The cells were seeded in 24-well plates; differentiation was induced using supplemented DMEM medium, dexamethasone (0.25 μM), IBMX (0.5 mM), and insulin (5 μg/mL). Once 80% confluence was reached, different concentrations of the extracts (100, 200, and 300 μg/mL) were added, and they were dissolved in DMSO (Dimethyl sulfoxide—used as a vehicle). At 0.05%, the medium was changed on the third day with supplemented DMEM medium and insulin (5 μg/mL) [32].

### 4.4. Cell Viability Determination Using MTT Assay

The cytotoxicity of BdHex, BdEAc, and BdHA was evaluated in a 3-(4,5-dimethylthiazol-2-yl)-2,5-diphenyltetrazol) bromide assay [33], also known as MTT. The 3T3-L1 cell line was cultured in 96-well plates and incubated at 5% CO_2_ atmosphere until reaching 80% confluency in DMEM + 10% FBS medium. The cells were added to a DMEM + SFB10% medium in addition to the different concentrations of the extracts (15–500 μg/mL), as well as 0.05% DMSO and 0.3–12.5 μg/mL paclitaxel (positive control) for 24 h at 37 °C and 5% CO_2_. After 24 h, the medium was removed, and the MTT dye (5 mg/mL in PBS) was placed in each well. The plates were sheltered from light and incubated for 4 h at 37 °C. The supernatant was removed, and 100 μL of isopropanol was added. After leaving the cultures were subjected to agitation for 15 min, readings were taken using a spectrophotometer at a wavelength of 490 nm. Cell viability percentages were obtained using the following equation:

Relative viability = (Control optical density − Sample optical density/Control optical density) × 100

Assays were performed in triplicate. Comparison values were made on the basis of 50% growth inhibition (IC50) in the cells treated with specific agents.

### 4.5. Quantification of Lipids Using Oily Red Assay

Lipid quantification was evaluated using an oily red assay (Sigma-Aldrich, according to the instructions for use, as well as microscopic visualization of the characteristics of lipid accumulation) on the 3T3-L1 cells induced for differentiation. 3T3-L1 pre-adipocytes were maintained in supplemented DMEM until reaching confluence, after which the medium was removed and the cells were fixed with 10% formalin for 20 min. The fixed cells were washed with an isopropanol–water solution, and subsequently, the oily red dye was placed on them. They remained covered and stirred for 25 min. They were then washed; the dye was recovered with isopropanol, and readings were made using a spectrophotometer at 490 nm.

### 4.6. Chemical Characterization of the Extracts (BdEAc and BdHA) for HPLC

Due to the activity found, a chemical analysis of the two most active extracts was carried out by means of HPLC. These analyses were performed using Waters brand equipment fitted with a Waters 2996 UV (900) photodiode array detector at 280 nm using Empower 3 software and a SUPELCOSIL packed column (Supelco, St. Louis, MO, USA. LC-F^®^, 25 cm 4.6 mm; 5 µm). The mobile phase consisted of 0.5% trifluoroacetic acid aqueous solution (solvent A) and acetonitrile (solvent B). The gradient system was as follows: 0–1 min, 0% B; 2–3 min, 5% B; 4–20 min, 30% B; 21–23 min, 50% B; 24–25 min, 80% B; 26–27 100% B; 28–30 min, 0% B. The flow rate was maintained at 0.9 mL min−1, and the sample injection volume was 10 μL. Absorbance was measured at 250 and 350 nm.

### 4.7. Fractionation of BdEAc

The ethyl acetate extract (BdEAc, 8 g) was absorbed in normal-phase silica (15 g, 70–230 mesh, Merck) and fractionated on a glass column packed with silica gel (80 g). For the mobile phase, a gradient of *n*-hexane/ethyl acetate was used with a 5% increase in polarity, collecting volumes of 100 mL, which were concentrated to obtain 118 fractions. The similarity of the compounds was analyzed using thin-layer chromatography (TLC), which allowed us to obtain 44 fractions (BdEAcR1 to BdEAcR44), which were grouped into four treatments for biological evaluation: TI from BdEAcR1 to BdEAcR6, TII from BdEAcR7 to BdEAcR20, TIII from BdEAcR21 to BdEAcR31, and TIV from BdEAcR32 to BdEAcR44; see Figure 8.

### 4.8. Obtaining the Triacetin, Kaempferol-3-O-rhamnoside, and Compounds (***3***–***5***) of Treatment III

The TIII treatment (300 mg) was adsorbed on silica gel (630 mg) and placed on a prepacked normal-phase plastic silica gel column (2 g), and a mobile-phase hexane/ethyl acetate/methanol mixture was used using four elution systems: 50:50, 50:50:5, 50:50:10, and 50.50:20 mL, respectively. Aliquots of 10 mL were obtained, giving a total of 24 fractions. Fraction TIII-8 was analyzed using ^1^H and ^13^C NMR, and triacetin (**1**, 2.3 mg) was identified as the major compound, with all of the following mass gases being identified: 2,3-dihydroxypropyl acetate (**2**), (3E)-2-methyl-4-(1,3,3-trimethyl-7-oxabicyclo[4.1.0]hept-2-yl)-3-buten-2-ol (**3**), 2,5-dihydroxyphenylacetic acid (**4**), and (3R)-3-hydroxydodecanoic acid (**5**). Fractions TIII-12 to TIII-24 (named TIII-B) (0.100 mg) were pooled and separated on a reverse-phase column (prepacked, Rp-18, 2 g), starting with 100% water and increasing polarity with 10% acetonitrile; 10 mL aliquots and 36 fractions were obtained. Fraction TIIIB-20 showed a single point in TLC, which was positive with the reagent for flavonoids (2-Aminoethyl diphenylborinate) and was identified as kaempferol-3-O-rhamnoside (**6**, 8 mg).

### 4.9. Obtention of Quercetin-3-O-rhamnoside from Treatment IV

BdEAcR32 to BdEAcR44 was carried out for purification. This junction was separated using open-column chromatography. Firstly, the column was packed with normal and reverse-phase silica, and mobile-phase water-acetonitrile was used as the system, starting with 7:3 water-acetonitrile. Samples that were 10 mL in volume and a total of 16 fractions were obtained. Fraction TIV-13 showed a single point in TLC, which was positive with the reagent for flavonoids (2-Aminoethyl diphenylborinate) and was identified as Quercetin-3-O-rhamnoside (**7**).

### 4.10. Quantitative Analyses of Kaempferol-3-rhamnoside and Quercetin 3-O-rhamnoside

The quantitative analyses of kaempferol-3-rhamnoside and quercetin 3-O-rhamnoside were carried out using calibration curves, which were separately constructed for the commercial standards using ascendant concentrations (6.5, 12.5, 25, 50, and 100 µg/mL), which were injected in triplicate. Kaempferol-3-rhamnoside and quercetin 3-O-rhamnoside produced the linear equations Y = 21,533X − 81,352; R^2^ = 0–99 and Y = 28,455X + 32,961; R^2^ = 0.99, respectively. The results of our quantitative analyses indicated that 1 g of BdEAc contained 9.8 mg of Kaempferol-3-o-rhamnoside and 6.31 mg of quercetin-3-O-rhamnoside.

### 4.11. GC-MS Analysis

Analyses of the chemical compositions of the TI-TIV treatments were carried out using gas chromatography–mass spectrometry using an Agilent Technology 6890 plus Gas Chromatograph that was coupled to a 5973N mass spectrometer with a simple quadrupole analyzer in an electron impact (IE) ionization mode at 70 eV [34]. The equipment contained an automatic injection system. Volatile and non-thermolabile organic compounds were separated on an HP 5MS capillary column (25 m long, 0.2 mm inner diameter, with a film thickness of 0.3 µm). The oven temperature was set at 40 °C for 2 min, then programmed from 40 to 260 °C at 10 °C/min and held for 20 min at 260 °C. The mass detector conditions were as follows: interface temperature—200 °C and mass acquisition range—20–550. The injector and detector temperatures were set at 250 and 280 °C, respectively. The splitless injection mode was used with 1 µL of each fraction (3 mg/mL solution). The carrier gas was helium, which was supplied at a flow rate of 1 mL/min. The volatile compounds were identified by comparing each mass spectrum with those of the National Institute of Standards and Technology (NIST) 1.7 Library.

### 4.12. Statistical Analysis

All experiments were performed three times. The differences between the groups were analyzed using a one-way ANOVA plus Dunnett’s post-test. The data are presented as the mean ± standard deviation, considering a = *p* < 0.05 and b = *p* < 0.001 to indicate statistical significance for the oily red lipid quantification tests.

## 5. Conclusions

Overweight and obesity are important concerns that increase the risk of many other health conditions. Both in Mexico and worldwide, they are considered risk factors for the development of chronic diseases such as diabetes mellitus II and hypertension. This has led to the need to continue looking for natural alternative treatments that could produce more clinical adherence.

The effect of extracts and chromatographic fractions from *Bauhinia divaricata* L. was evident in this pharmacological model, and the major compounds in the most active treatments could be responsible for the antiadipogenic activity in the TIII and TIV mixtures, such as triacetin, kaempferol-3-O-rhamnoside, and quercetin 3-O-rhamnoside. Therefore, we can conclude that the presence of these compounds is essential for the inhibition of the differentiation of preadipocyte-like cells into mature adipocytes. This knowledge is very useful for proposing the development of potential medications against obesity or overweight, which can be developed using standardized extracts rich in these polyphenolic compounds.

## Figures and Tables

**Figure 1 plants-12-03799-f001:**
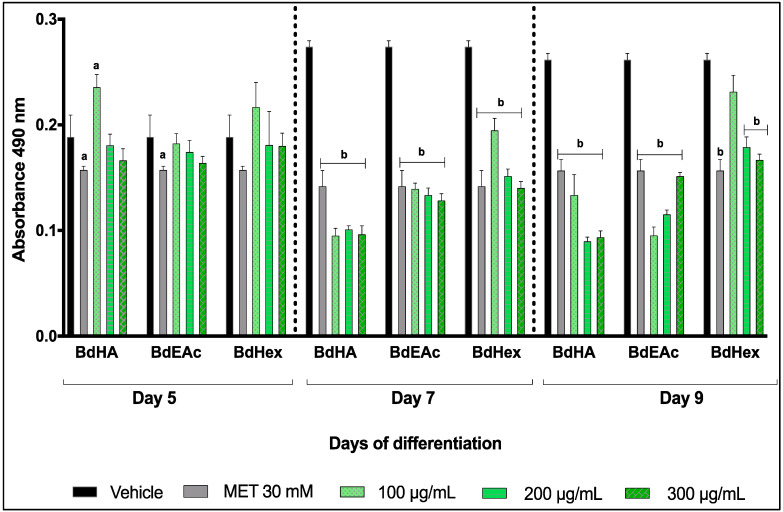
Quantification of lipids in 3T3-L1 cells treated with different extracts of *Bauhinia divaricata*. Readings were performed on days 5, 7, and 9 of differentiation. A one-way ANOVA plus Dunnett’s post-test was performed with respect to the vehicles. a = *p* < 0.05 and b = *p* < 0.001 indicate statistical significance. Met = metformin.

**Figure 2 plants-12-03799-f002:**
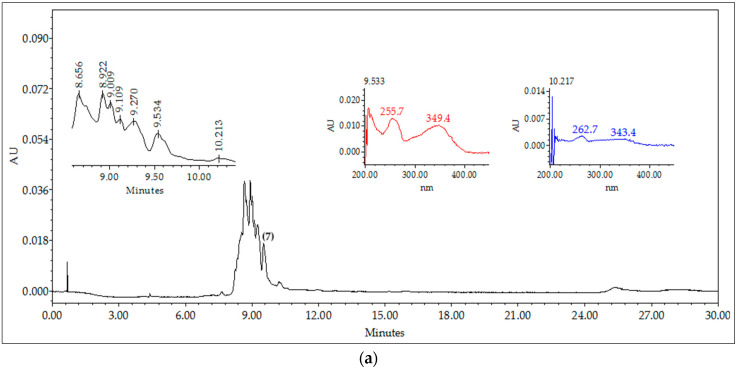
HPLC chromatograms and UV spectra: (**a**) hydroalcoholic extract (BdHA) 9.5 (quercetin 3-O-rhamnoside (**7**) and (**b**) fraction (BdEAc) 9.5 (quercetin 3-O-rhamnoside (**7**)), 10.2 (kaempferol-3-O-rhamnoside (**6**)). The chromatographic fingerprint of both extracts indicates the presence of flavonoid-type compounds and phenols. Empower 3 software and a SUPELCOSIL-packed column (Supelco, St. Louis, MO, USA. LC-F^®^, 25 cm; 4.6 mm; 5 µm). The mobile phase consisted of 0.5% trifluoroacetic acid aqueous solution (solvent A) and acetonitrile (solvent B).

**Figure 3 plants-12-03799-f003:**
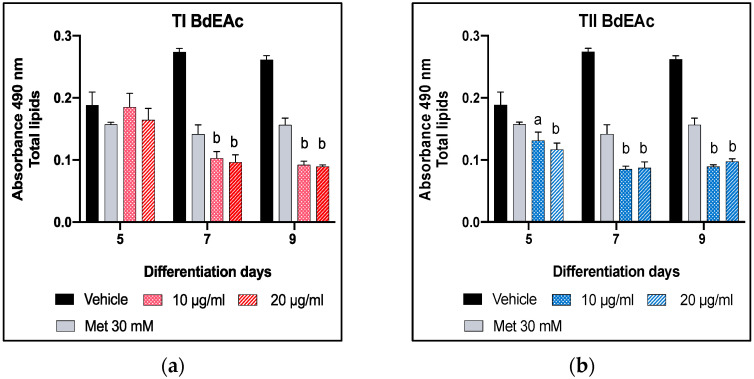
Quantification of lipids in 3T3-L1 cells treated with TI = from BdEAcR1 to BdEAcR6 (**a**), TII = from BdEAcR7 to BdEAcR20 (**b**), TIII = from BdEAcR21 to BdEAcR31 (**c**), and TIV = from BdEAcR32 to BdEAcR44 (**d**). Readings were performed on days 5, 7, and 9 of differentiation. A one-way ANOVA plus Dunnett’s post-test was performed with respect to vehicles. a = *p* < 0.05 and b = *p* < 0.001 indicate statistical significance. Met = metformin.

**Figure 4 plants-12-03799-f004:**
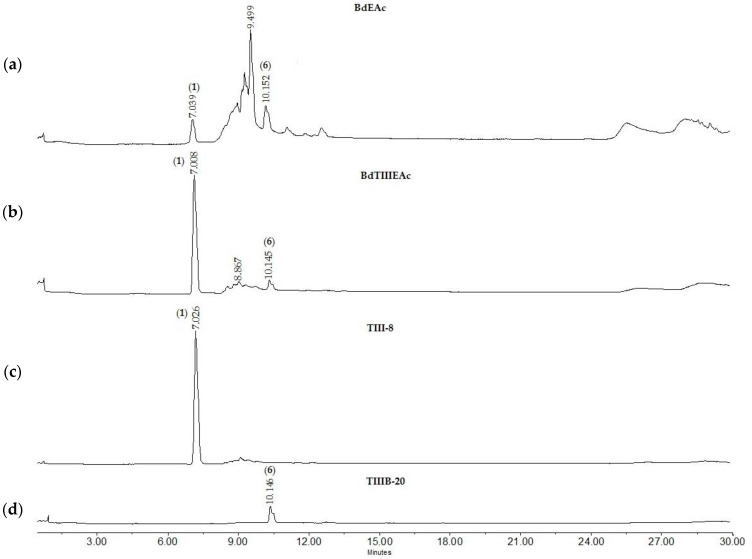
Chromatographic profiles of treatment (**a**) BdEAc, (**b**) BdTIIIEAc, (**c**) TIII-8, and (**d**) TIIIB-20.

**Figure 5 plants-12-03799-f005:**
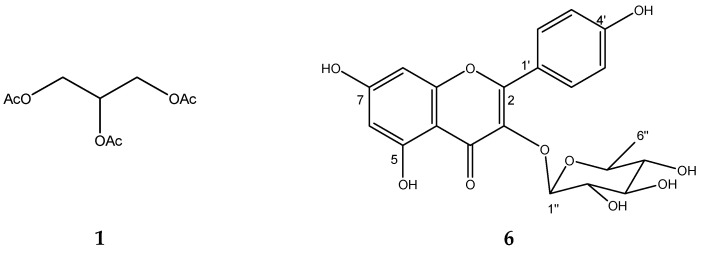
Chemical structure of triacetin (**1**) and kaempferol-3-O-rhamnoside (**6**) from TIIIBdEAc.

**Figure 6 plants-12-03799-f006:**
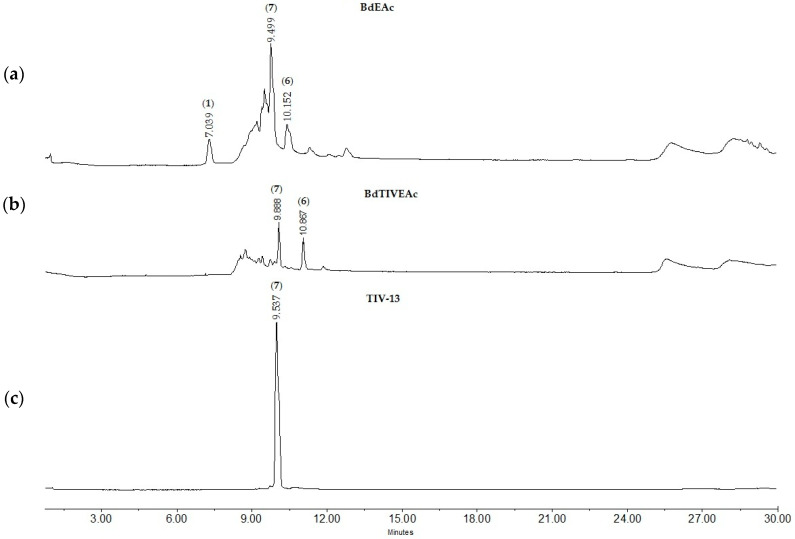
Chromatographic profiles of treatments: (**a**) BdEAc, (**b**) BdTIVEAc, and (**c**) TIV-13.

**Figure 7 plants-12-03799-f007:**
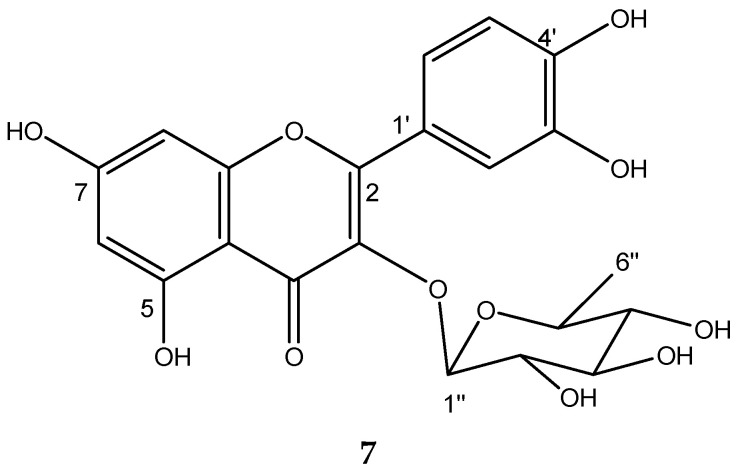
Chemical structure of quercetin 3-O-rhamnoside (**7**) from TIV-13.

**Figure 8 plants-12-03799-f008:**
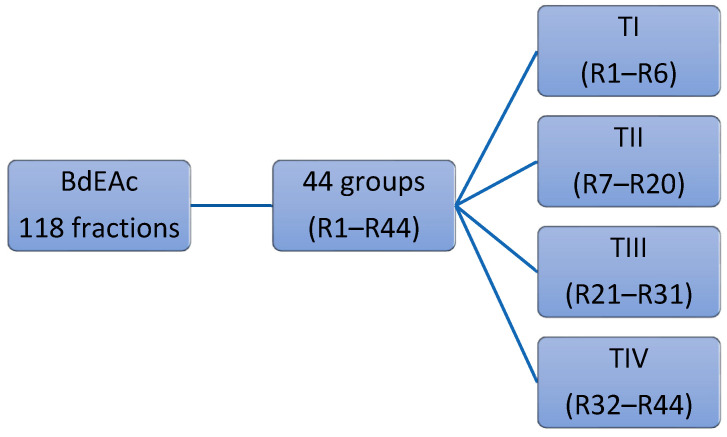
The chromatographic strategy of ethyl acetate fraction (BdEAc).

**Table 1 plants-12-03799-t001:** Cell viability from *B. divaricata* extracts.

Extracto	IC_50_ (μg/mL)
BdHex	>1000
BdEAc	>1000
BdHA	>1000
Paclitaxel (+)	14.32

(BdHex) = hexanic extract; (BdEAc) = ethyl acetate extract; (BdHA) = hydroalcoholic extract and (+) positive control.

**Table 2 plants-12-03799-t002:** Spectroscopic data of NMR of triacetin (**1**) (600 MHz, CDCl_3_, δ ppm).

Position	^1^H (δ, J in Hz) 1	*δ*^13^C(δ) 1
**1a** **b**	4.14, (dd, 6.3, 11.7) 4.19, dd (4.5, 11.6)	65.52
**2**	3.93 (m)	70.32
**3a** **b**	3.70 (dd,3.9, 11.5) 3.60 (dd, 5.9, 11.5)	63.45
**4**	-----------------	171.6
**5**	2.1 (s)	20.98

**Table 3 plants-12-03799-t003:** Comparison of spectroscopic data of ^1^H and ^13^C NMR of kaempferol-3-O-rhamnoside (**6**, 600 MHz, DMSO) with the relevant data in the literature.

			Amer, A. A et al., 2022. [18]		Elloumi, W et al., 2022. [19]	
Position	^1^H (δ, J in Hz) 6	*δ*^13^C(δ) 6	^1^H (δ, J in Hz) 6	*δ*^13^C(δ) 6	^1^H (δ, J in Hz) 6	*δ*^13^C(δ) 6
2	---	160.2	---	157.2	---	---
3	---	134.7	---	---	---	---
4	---	178.1	---	178.5	---	---
5	---	164.1	---	162.2	---	---
6	6.21 (1H, d, 1.9)	98.6	6.25 (1H, d, 1.8)	---	6.19 (s)	---
7	---	157.2	---	164.2	---	---
8	6.38 (1H, d, 1.9)	93.4	6.45 (1H, d, 2.3)	---	6.36 (s)	---
9	---	157.8	---	160.1	---	---
10	---	104.3	---	---	---	---
1′	---	122.8	---	---	---	---
2′	7.78 (1H, d, 8.7)	130.4	7.84 (2H, dd, 8.6)	---	7.76 (d, 8.4)	---
3′	6.95 (1H, d, 8.7)	115.1	7 (2H, dd, 8.6)	---	6.93 (d, 8.4)	---
4′	---	161.8	---	157.6	---	---
5′	6.95 (1H, d, 8.7)	115.1	7 (2H, dd, 8.6)	---	6.93 (d, 8.4	---
6′	7.78 (1H, d, 8.7)	130.4	7 (2H, dd, 8.6)	---	7.76 (d, 8.4)	---
1′′	5.39 (1H, d, 1.4)	102.1	5.52 (1H, d,1.4)	---	5.38 (d, 1.5)	---
2′′	4.23, (1H, dd, 1.4, 3.3)	70.52	4.22 (1H, d, 1.4)	---	4.23 (dd, 3.3, 1.7)	---
3′′	3.72, (1H, dd, 3.3, 8.8)	70.72	3.70	---	3.72 (m)	---
4′′	3.32, (m)	71.7	---	3.30	3.34 (m)	---
5′′	3.32, (m)	70.6	---	---	3.34 (m)	---
6′′	0.94, (1H, d, 5.5)	16.2	---	0.90 (3H, d, 6.0)	0.93 (d, 5.7)	---

**Table 4 plants-12-03799-t004:** Data from the gas chromatography–mass spectrometry analysis of TIII-8.

RT (min)	Area(%)	MolecularWeight(g/mol)	Compound Name	Chemical Structure
9.356	68.89	134.13	2,3-Dihydroxypropyl acetate	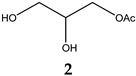
14.138	3.87	218.2	Triacetin	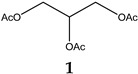
14.782	9.12%	216.32	(3R)-3-Hydroxydodecanoic acid	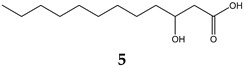
15.353	3.19%	168.15	2,5-Dihydroxyphenylacetic acid	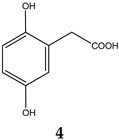
17.304	2%	224.34	(3E)-2-Methyl-4-(1,3,3-trimethyl-7-oxabicyclo[4.1.0]hept-2-yl)-3-buten-2-ol	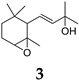

## Data Availability

Not applicable.

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
