# Peer review of "Pharmacological and Chemical Analysis of Bauhinia divaricata L. Using an In Vitro Antiadipogenic Model"

_plants, 2023, doi:10.3390/plants12223799_

Round 1

Reviewer 1 Report

Comments and Suggestions for Authors

In the current paper by  Laura Islas-Garduño et al.  lipolytic and antiadipogenic activity of Bauhinia divaricata L. in 3T3-L1 cells Several extracts hexanic (BdHex), ethyl acetate (BdEAc) and hydroalcoholic (BdHA), extracts were prepared. Lipid levels were measured in 3T3-L1 cells differentiated into adipocytes. The evaluation of the cell viability identified an IC50 >1000 μg/ml, in all the extracts and the evaluation of the antiadipogenic activity indicated that there is a significant reduction (p<0.001) in the accumulation of lipids with the extracts hydroalcoholic (60%) and ethyl acetate (75%) of B. divaricate compared with metformine at 30 mM (65%). The major compounds identified in these extracts were: triacetin (1), 2,3-dihydroxypropyl acetate (2), (3E)-2-methyl-4-(1,3,3-trimethyl-7-oxabicyclo[4.1.0]hept-2-yl)-3-buten-2-ol (3), 2,5-dihydroxyphenylacetic acid (4), (3R)-3- hydroxydodecanoic acid (5), kaempferol-3-O-rhamnoside (6) and quercetin 3-O-rhamnoside (7). Comments paper is sound isolation of compounds and biological tests performed. 

1. Please check English language and nutritional composition

2. Please add HPLC MS ANALYSES OF THE ALL EXTRACTS that can be a good plus. And the established HPLC systems can be good fingerprint for quality control purposes.

3. Please quantify by HPLC MS OR DAD the compounds in the extracts 

Comments on the Quality of English Language

Need to improve

Reviewer 2 Report

Comments and Suggestions for Authors

Please, see the file attached.

Comments on the Quality of English Language

Moderate editing of the English language is required.

Reviewer 3 Report

Comments and Suggestions for Authors

In this manuscript (plants-2676317) entitled "Pharmacological and chemical analysis of Bauhinia divaricata using an in vitro antiadipogenic model" submitted to Plants, Ana Laura Islas-Garduño and colleagues have evaluated the lipolytic and antiadipogenic activity of Bauhinia divaricata L. in 3T3-L1 cells and identified the major compounds in the bioactive treatments. This research is interesting and convincing, but minor points need to be addressed to improve the quality of this manuscript.

1. For Figures 1 and 3, authors showed quantification of lipids by absorbance 520nm. At the section of “Materials and Methods”, quantification of lipids by oily red assay was performed by spectrophotometer 492 nm, which should be explained in the revision.

2. For Figures 1 and 3, the reading was performed on days 5, 7 and 9 of differentiation, what is the situation for the day 0 of differentiation.

3, The Section of “Introduction” is too simple. Previous studies on pharmacological and physilogical analysis of Bauhinia divaricate should be mentioned in the revised Introduction.

4, For Line 14, full name of the abbreviation BMI should be spelt out at their first appearance in the revised manuscript.

5, For Line 301, 4.33 is incorrect.

Reviewer 4 Report

Comments and Suggestions for Authors

Authors of this manuscript entitled “Pharmacological and chemical analysis of Bauhinia divaricata using an in vitro antiadipogenic model” studied lipolytic and antiadipogenic activity of different extracts of Bauhinia divaricata L. and isolated various known phytomolecules such as triacetin, (3R)-3- 26 hydroxydodecanoic acid, kaempferol-3-O-rhamnoside and quercetin 3-O-rhamnoside. Overall, the manuscript is not presented well and experimental plan and important data from the manuscript is missing. My feedback on this manuscript is as follows:

            -Abstract is not good, its too long and should be rewritten avoiding experimental detail and introductory remarks.

-Abstract should be concise, informative and to the point.

-The paper requires several English grammar and spelling corrections.

-The introduction is brief; it would be beneficial if the author could expand it by explaining importance of natural products and including some new references such as.

  • DOI: 10.1007/s10600-008-9090-3

·         DOI: 10.1007/s10600-008-9093-0

·         https://doi.org/10.1002/hlca.200890256

Authors should provide scope of this study. Given recent advancements in the field.

-HPLC Chromatograms are blurred and must be improved.

-Chromatogram for GC analysis should be included

-In chromatograms (GC and HPLC), authors should mention the name of the identified compounds for their respective peak.

-bioactivity of isolated compounds:  why authors did not test the isolated compounds for the bioactivity.

-Reporting activity at extract level is not informative when the plant is already known for said bioactivity. It is imperative to test the isolated compounds.

-Lack of experimental detail. GC analysis methodology is not included in the manuscript, why???

- Lack of Conclusion: The conclusion does not summarize the deliverables or the implications of the results. It is essential to provide a synthesis of the key perspectives and their significance and must be wrap up with concluding remarks.

Comments on the Quality of English Language

Moderate editing of English language required

Round 2

Reviewer 1 Report

Comments and Suggestions for Authors

paper was substantially ameliorated and can now be published

Author Response

We appreciate the revision process as well as the final comments.

Comments paper is sound isolation of compounds and biological tests performed.

COMMENTS:
Paper was substantially ameliorated and can now be published.

RESPONSES:
We appreciate the revision as well as your final comments.

Reviewer 2 Report

Comments and Suggestions for Authors

The authors have complied with the reviewer's recommendations.

Author Response

We aprecciate the revision process as well as the final comments.

The present work addresses a very important aspect of human health - obesity and excessive fat accumulation. I have the following recommendations that will contribute to the better appearance of this article.

COMMENTS
 The authors have complied with the reviewer's recommendations.

RESPONSES
We appreciate the revision well as your final comments.

Reviewer 4 Report

Comments and Suggestions for Authors

Authors have not improved the manuscript and their response to the queries are not satisfactory, some of the major queries which authors did not responded are given below as example:

-Chromatogram for GC analysis has not been included

- Chromatograms have not been improved and names of the identified compounds for their respective peaks are not included in revised Manuscript

-Lack of experimental detail. GC analysis methodology is not complete

- Lack of Conclusion: This section still not good and lack concluding remarks and scope of the study, contain English typographical mistakes

-There are still so many English grammar and typographical mistakes through out the manuscript.

Authors have not revised the manuscript sincerely and I will leave the decision on Editors.

Comments on the Quality of English Language

Moderate editing of English language required

Author Response

We appreciate the revision process, our responses wereplaced in the attached file. 
